# High-Payload Data-Hiding Method for AMBTC Decompressed Images

**DOI:** 10.3390/e22020145

**Published:** 2020-01-25

**Authors:** Jung-Yao Yeh, Chih-Cheng Chen, Po-Liang Liu, Ying-Hsuan Huang

**Affiliations:** 1Graduate Institute of Precision Engineering, National Chung Hsing University, No. 250, Kuo-Kuang Road, Taichung City 402, Taiwan; honor0425@gmail.com; 2Department of Computer Science and Engineering, National Chung Hsing University, No. 250, Kuo-Kuang Road, Taichung City 402, Taiwan; salu.chen@gmail.com; 3Aeronautical Systems Research Division, National Chung-Shan Institute of Science and Technology, Taichung 40722, Taiwan; ying.hsuan0909@gmail.com

**Keywords:** data hiding, AMBTC, steganography, stego image, dictionary-based coding, pixel value adjusting

## Abstract

Data hiding is the art of embedding data into a cover image without any perceptual distortion of the cover image. Moreover, data hiding is a very crucial research topic in information security because it can be used for various applications. In this study, we proposed a high-capacity data-hiding scheme for absolute moment block truncation coding (AMBTC) decompressed images. We statistically analyzed the composition of the secret data string and developed a unique encoding and decoding dictionary search for adjusting pixel values. The dictionary was used in the embedding and extraction stages. The dictionary provides high data-hiding capacity because the secret data was compressed using dictionary-based coding. The experimental results of this study reveal that the proposed scheme is better than the existing schemes, with respect to the data-hiding capacity and visual quality.

## 1. Introduction

The concealment of information within media files is commonly used in various applications. This process originates from the hieroglyphs used in the Egyptian civilization. Other cultures, such as the Chinese culture, adopted a more physical approach to hide messages by writing them on silk or paper, rolling the material into a ball, and covering the material with wax to communicate political or military secrets. Data hiding is nearly indispensable for every aspect in our daily lives whether for good or evil intentions.

Due to its rapid growth, the Internet has recently become far more popular than traditional media. Data is accessible by everyone due to the popularity of the Internet. Therefore, possessing the capabilities of detecting copyright violations, forgery, and fraud is crucial. Many techniques, such as steganography and cryptography, have been designed to secure digital data. The difference between steganography and cryptography is as follows: In cryptography (e.g., chaos-based encrypted systems, secure pseudo-random number generator, etc. [1]) users are aware that there is an encrypted image, but they cannot efficiently decode the encrypted image unless they know the proper key. In steganography, users can easily decode the encrypted message, but most people do not notice that there is an encrypted message. In this study, we focused on the techniques used for hiding data in images.

The schemes present for hiding data in an image can be broadly classified into two categories, irreversible data-hiding schemes [2,3,4] and reversible data-hiding schemes [5,6,7]. In the irreversible data-hiding schemes, a recipient can extract the secret information. However, the original image cannot be recovered after extracting the secret information. In the reversible data-hiding schemes, the hidden data can be extracted from the image, and the original image can be retrieved from a stego image without any distortion. Two factors affect a data-hiding scheme, i.e., visual quality and embedding payload. A high-quality data-hiding scheme should not raise any suspicions of adversaries. Therefore, this type of scheme should provide low image distortion and high payload.

To decrease the size of a digital image file or accelerate the transmission, a data-hiding scheme that employs a compressed image should be developed. Many compressed file formats have been proposed, such as JPEG and JPEG2000. Wang et al. [8] proposed a lossless data-hiding method for JPEG images by using adaptive embedding. Lee et al. [9] proposed a scheme in which a secret image was compressed using JPEG2000 and then, embedded in the cover image by using tri-way pixel value differencing. Nevertheless, both JPEG and JPEG2000 need complicated computation for image compression and decompression.

Another popular technique used for image compression is block truncation coding (BTC) [10]. Compared with the methods using JPEG and JPEG2000, BTC is a simple and efficient encoding technique that is used for image compression. Therefore, the computation cost is relatively low when a data-hiding scheme is based on BTC. 

Lema and Mitchell [11] proposed the absolute moment BTC (AMBTC) technique to improve the compression performance of BTC. When AMBTC is used, the first absolute moment is maintained with the mean. To exploit the advantages of AMBTC compression, we proposed an AMBTC decompressed image-based data-hiding scheme by using a pixel adjusting strategy.

The basic idea of the proposed study is to preliminarily calculate the probability of secret data and then select the best codebook for embedding the secret data. The secret data are embedded into the AMBTC compression image by modifying the pixel value according to the codebook. Experimental results reveal that the proposed scheme is almost better than the current state of the art method in terms of the hiding capacity.

The remainder of the paper is organized as follows: Section 2 describes the relevant approaches such as the BTC and AMBTC techniques for data hiding; Section 3 describes the implementation flow of the scheme proposed for data hiding; Section 4 discusses several experimental results are presented, and some issues; and finally, Section 5 specifies the conclusions and future work.

## 2. Related Works 

Before describing the high data-hiding capacity of the proposed scheme, we review the AMBTC technique and some recently developed AMBTC-based data-hiding methods.

### 2.1. Absolute Moment Block Truncation Coding (AMBTC)

BTC, a simple and efficient block-based lossy image compression method, is used for grayscale images. Although the BTC method provides a low compression ratio, it is a popular image compression method because of its low complexity with respect to both computation and implementation. In the BTC algorithm, an image X, with M × N pixels, is divided into nonoverlapping blocks. Each block has n × n pixels, and the pixel values can be different. The mean and standard deviation of each pixel value are calculated before conducting BTC. In general, two statistical characteristics change from one block to another.

The hardware implementation of BTC is challenging because the square and square root functions are involved. To resolve this problem, AMBTC [11] was proposed as a type of BTC. The AMBTC uses the first absolute moment and mean values instead of using the standard deviation value. The main difference between AMBTC and BTC is that the mean and standard deviation values of a block are preserved in BTC. However, in AMBTC, the high mean and low mean values of a block are preserved.

As in BTC, an image X is divided into nonoverlapping blocks with n × n pixels also in the AMBTC encoding phase. For each block, the mean $\bar{x}$ and the absolute moment α of the pixel values are calculated using
(1)$$\bar{x}=\frac{1}{m}\sum_{i=1}^{m}x_{i},$$
(2)$$\alpha =\frac{1}{m}\sum_{i=1}^{m}|x_{i}-\bar{x}|.$$
Note that m = n × n.

The pixel value $x_{i}$ is compared with the mean $\bar{x}$ for composing a bit plane for each pixel in the block. If the pixel value $x_{i}$ is greater than the mean $\bar{x}$, then $x_{i}$ is denoted as 1. Otherwise, the pixel value is denoted as 0. The equation of bit representation is
(3)$$P_{i}=\left\{ \begin{array}{l} 1,\;\mathrm{if}\;x_{i}>\bar{x},\\ 0,\;\mathrm{otherwise}. \end{array}\right.$$

In the AMBTC-compressed block reconstruction phase, the block reconstruction is conducted using two values $L_{m}$ and $H_{m}$. The values of $L_{m}$ and $H_{m}$ are computed using
(4)$$L_{m}=\bar{x}-\frac{ma}{2(m-q)},$$
(5)$$H_{m}=\bar{x}+\frac{ma}{2q}.$$

In Equations (4) and (5), q represents the number of pixels with pixel values greater than $\bar{x}$. Thus, a compressed block has two values $L_{m}$ and $H_{m}$, where $L_{m}$ is the low mean value and $H_{m}$ is the high mean value. To reconstruct a block, the pixels that are assigned the value of 0 in the bit plane are replaced with the $L_{m}$ value, and the pixels assigned the value of 1 in the bit plane are replaced with the $H_{m}$ value by
(6)$${x^{\prime }}_{i}=\left\{ \begin{array}{l} H_{m},\;\mathrm{if}\;p_{i}=1,\\ L_{m},\;\mathrm{if}\;p_{i}=0. \end{array}\right.$$

### 2.2. Related Work of BTC and AMBTC Based Data Hiding Schemes

BTC has significantly low complexity and requires less memory. Therefore, BTC is a good scheme for data hiding. Chuang and Chang proposed a data-hiding scheme for BTC-compressed images for embedding data in the bitmaps of smooth blocks to obtain an improved image quality. There are two steps in in the embedding process of the scheme proposed by Chuang and Chang. Initially, a cover image is compressed into blocks by using BTC for calculating two quantized data and the bit plane corresponding to each block. Finally, the secret data is embedded into the bitmaps of the predefined smooth blocks that satisfy the following equation: $H_{m}-L_{m}<Threshold$. The smooth blocks were selected because bit replacement in these bit planes causes a slight distortion in the BTC image. In the extraction process of the scheme proposed by Chuang and Chang [12], the difference $H_{m}-L_{m}$ has to be first calculated. If $H_{m}-L_{m}<Threshold$, then the secret bit in the bit plane *p*′*_i_* is extracted. However, in this scheme, the stego image quality degrades significantly as the threshold values increases.

Hong et al. [13] proposed a reversible data-hiding scheme based on bit plane flipping according to the corresponding secret bit. In the embedding process, each image block was compressed using AMBTC-compressed codes to determine whether the block is embeddable or not. If $L_{m}<H_{m}$, then the block is considered embeddable. Otherwise, the block is considered non-embeddable. For each embeddable block, if the secret bit is 1, then the bit plane $p_{i}$ is flipped to ${\bar{p}}_{i}$, where ${\bar{p}}_{i}$ is not an operator. If the secret bit is 0, then no operation is required. In the extraction process, if $L_{m}>H_{m}$ in *p*′*_i_*, then the secret in *p*′*_i_* is 1. Otherwise, the secret bit in *p*′*_i_* is 0. The scheme presented by Hong et al. does not hide data in blocks with $L_{m}=H_{m}$. Therefore, Chen et al. [14] proposed a reversible data-hiding method to improve the scheme by Hong et al. The AMBTC-compressed block that has $L_{m}=H_{m}$ is a smooth area, which is considered unnecessary bit plane information. Thus, the secret bit can be embedded all bits in the bit plane block to improve the scheme by Hong et al.

Li et al. [15] introduced a data-hiding scheme by using the histogram shifting technique on BTC-compressed mean tables for further improving the hiding capacity, while maintaining the quality of the BTC-compressed image. The hiding scheme comprises two main steps. The first step is based on the bit plane flipping method that hides secret bits by swapping the high mean and low mean values. In the second step, histogram shifting is conducted on the resulting mean tables after swapping. This scheme requires no additional data in the stego code stream. Therefore, very low distortion is observed in this scheme after data embedding, and the security of the embedded data is enhanced. However, this technique cannot provide a sufficient data-hiding capacity and requires overhead information to record a histogram.

Lin et al. [16] proposed a technique to explore the redundancy in a block of AMBTC-compressed images to determine whether the block is embeddable. If the secret bits and bit plane combined in the block has more than three different cases, the block is marked as an embeddable block. Four disjoint sets were created using this technique of embeddable blocks for embedding data using different combinations of the mean value and its standard deviation.

Ou and Sun [17] proposed a data-hiding scheme with minimum distortion based on AMBTC. In this scheme, a predefined threshold is used to determine if a block of the AMBTC-compressed codes is a smooth or complex block in which data are embedded. If an AMBTC-compressed block $H_{m}-L_{m}<Threshold$, then the block is considered a smooth block. All bit planes in smooth blocks are used to embed data by replacing the bits of the block with secret data bits. The two quantization levels in the smooth block are then recalculated to reduce distortion in the image. In the complex blocks, a proportion of secret bits were concealed by exchanging the order of two quantization levels and toggling the bit plane. By performing this method, the payload can be increased without any distortion. Both smooth and complex blocks can be used to embed data in an AMBTC-compressed block. Therefore, the payload of this scheme was obviously enhanced. 

Malik et al. [18] modified the AMBTC compression technique for embedding secret data. In their method, one-bit plane is converted to two-bit planes that can attain better image quality and high capacity. Although this scheme has high visual quality and high payload, it causes permanent distortion to the original AMBTC code and requires overhead information. Malik et al. [19] proposed an AMBTC compression-based data-hiding scheme by using the pixel value adjusting strategy. In this technique, the stream of secret bits was converted to digits with a base of three. Then, the pixel values of the AMBTC-compressed block are modified, at the most by one, to hide secret data. This scheme could maintain a balance between the hiding capacity and quality of a stego image.

As discussed above, data hiding by using the AMBTC technique is an issue worthy of more research. In this study, we extended the work of Malik et al. [19] to embed a larger amount of secret data. In the next section, the proposed scheme is discussed.

## 3. Proposed Scheme

Figure 1 shows the flowchart of our application. First, one monitoring image on the unmanned aerial vehicle was compressed because the transmitting volume of wireless network is limited. When the command post or chief’s car receives the compression codes, they are decoded as the decompressed image. In addition, they embed secret data into the reconstructed image, thereby cheating hackers and avoiding attacks. Finally, the headquarters can extract secret data and recover the decompressed image.

The main aim of the study is to present a data-hiding scheme with high data-hiding capacity and high image quality. In the scheme, secret data is hidden in an AMBTC decompressed image. The AMBTC decompressed image is losslessly reconstructed and the secret data, then, is losslessly revealed from the reconstructed image. The AMBTC encoding procedures are described in Section 2. Before embedding the secret data, the cover image must be compressed using the AMBTC algorithm. In other words, the proposed scheme uses the AMBTC decompressed image to embed the secret data.

The proposed scheme involves three stages: In the first stage, an appropriate encoding and decoding dictionary is found. The dictionary is used in the second stage to embed the data. In the third stage, the secret data is extracted. The details of the proposed scheme are presented in Figure 1.

### 3.1. Finding a Unique Decodable Dictionary

A binary secret sequence S comprises 0 and 1 values and is denoted as *S*
$=\left\{s_{1},\;s_{2},\ldots ,s_{N}\right\}$, where $s_{i}\in \left\{0,1\right\}$ for i=1~N. Consider the dictionaries *D_1_*, *D_2_*, *D_3_*, and *D_4_* formed using K subsets of S, that is, $S_{p1},S_{p2},\ldots ,\;S_{pK}$. Different image quality is obtained due to the different dictionaries. Thus, we can calculate each probability of symbol $S_{p}$ in S. The amount of information in each symbol $I_{a}$ can be represented by
(7)$$I_{a}=-log_{2}\left(\mathrm{pr}\left(S_{pk}\right)\right).$$

Then, the average information per symbol interval is $H\left(S_{pk}\right)$ and can be represented by
(8)$$H\left(S_{pk}\right)=-\overset{n}{\underset{k=1}{\sum }}pr(S_{pk})log_{2}\left(pr(S_{pk}\right)).$$

The average information $H\left(S_{pk}\right)$ is referred to as the entropy. The dictionary with the smallest entropy *H* should be selected because it can achieve the best encoding benefit. The following explains why the dictionary of the smallest entropy is used: Assume that there is only one symbol’s type in the whole secret sequence. In other words, the other types never occur. In this case, the entropy is equal to 0, i.e., $H\left(S_{pk}\right)=0.$ Afterwards, the specific symbols are replaced by the absolute minimum value “0”, thereby controlling the distortion level in the data embedding phase. Consequently, the proposed method selects the dictionary of the smallest dictionary.

An example is used to explain the above procedure. Assume the secret sequence S={001110111100110110010000010011010}. In dictionary $D_{1}$ listed in Table 1, the secret sequence is represented as S={001,11,01,11,10,01,10,11,001,000,001,001,10,10} for easy readability. According to $D_{1}$, the total number of information is 12.4670 and the average information $\;H\left(S_{pk}\right)$ per symbol at S is 2.1570. In dictionary $D_{2}$, which is listed in Table 2, the secret sequence can be represented as S={00,11,10,11,11,00,11,011,00,10,00,00,10,011,010}. According to $D_{2}$, the total number of information is 12.1451 and the average information$\;H\left(S_{pk}\right)$ per symbol at S is 2.2264. The third and fourth dictionaries are constructed in the same manner, and their entropies values are listed in Table 3 and Table 4, respectively. Obviously, the entropy of $D_{1}$ is the smallest among all the dictionaries. Therefore, we used $D_{1}$ to encode the secret sequence.

Subsequently, the symbols in the selected dictionary are encoded further to obtaining the embedded digits. According to the rule of thumb of data encoding, $S_{p}$ with the maximum occurrence frequency was encoded as the absolute minimum value. By contrast, $S_{p}$ with the lowest occurrence frequency was encoded as the absolute maximum value. Consequently, $S_{p}$ was sorted based on the occurrence frequency, and then, its sorted index was encoded to obtain the adjusting pixel values $P_{v}$, i.e.,
(9)$$p_{v}=\left\{ \begin{array}{l} -\left\lfloor \frac{Sort\;index}{2}\right\rfloor ,\;\mathrm{if}\;\mathrm{Sort}\;\mathrm{index}\;\mathrm{is}\;\mathrm{an}\;\mathrm{odd}\;\mathrm{number},\\ \left\lfloor \frac{Sort\;index}{2}\right\rfloor ,\;\mathrm{otherwise}. \end{array}\right.$$

The following example is used to explain how to encode most symbols as smaller digits, as listed in Table 5. The occurrence frequencies of two symbols, “001” and “10”, are 4, which are higher than those of other symbols. According to Equation (9), the symbol “001” is encoded as the absolute minimum value “0”. Moreover, the symbol “10” is encoded as the second smallest value “1”. The remaining symbols are encoded in the same manner.

### 3.2. Embedding Stage

The AMBTC decompressed blocks $b_{i}$ in the original AMBTC decompressed image T are sequentially scanned. If the difference between $H_{m}$ and $L_{m}$ is smaller than 4, then the block is considered a non-embeddable block. Otherwise, the block is an embeddable block. In the first embeddable block, the binary representation of the ID number of the selected dictionary is embedded into the least significant bits (LSBs) of the second Hm and the second $L_{m}$. Note that the number of dictionaries is four, thus the two LSBS can effectively represent the ID number. The other blocks are then used to embed the secret data by using the pixel value adjusting strategy.

In each embeddable block, the first $H_{m}$ and the first $L_{m}$ are defined as non-embeddable pixels, which are used as the reference information of data extraction and image recovery. For the embeddable block $b_{i}$, each pixel $x\prime_{i}$ except the first $H_{m}$ and the first $L_{m}$ is increased by the adjusting pixel values $P_{v}$, that is, ${x\prime \prime }_{i}=x\prime_{i}+P_{v}$. The difference between maximum $P_{v}$ and minimum $P_{v}$ in the difference D is equal to 4. It implies that the distortion of pixels is low. The embedding pseudocode is shown in Algorithm 1 as follows:
**Algorithm 1:** Embedding pseudocode$\mathrm{foreach}\;AMBTC-compressed\;block\;b_{i}in\;T\;do$    $\mathrm{if}\;H_{m}-L_{m}\leq 4$**then**  /* non-embeddable block */
      Do nothing;    $else\;if\;b_{i}\;is\;first\;embeddable\;block\;then$      embedding dictionary D number;    else       $foreach\;pixel\;x\prime_{i}\;in\;b_{i}\;do$       $\mathrm{find}\;\mathrm{adjusting}\;\mathrm{pixel}\;\mathrm{values}\;P_{v}\;\mathrm{in}\;D;$       $x\prime \prime_{i}=x\prime_{i}+\;P_{v}$;      end    end  end

Figure 2 displays the embedding example in which ={0011101111001101100
10000010011010}. Figure 2a presents the appropriate dictionary D found in Section 3.1. This dictionary was used to encode the secret sequence. After looking up the dictionary D, S is divided into many subsets $S_{p}$, as shown in Figure 2b. These subsets are mapped using the adjusting pixel values $P_{v}$, which are just the embedded value.

To embed these values, the original block must be compressed and decompressed using the AMBTC algorithm, as shown in Figure 2c. After using the AMBTC algorithm, the AMBTC decompressed block can be reconstructed using a low mean value $L_{m}$ of 97 and a high mean value $H_{m}$ of 155, as shown in Figure 2d. Both the first $L_{m}$ and first $H_{m}$ are non-embeddable pixels and are marked with yellow color for easy readability. They are used as reference information of data extraction and image recovery. For the AMBTC decompressed block, the pixel, except the first $H_{m}$ and the first $L_{m}$, is increased by the adjusting pixel values $P_{v}$ to obtain the stego pixel. Figure 2e shows the stego block.

If the overflow or underflow problem occurs in any altered pixel of the block, then all of the pixels in the corresponding block remain unchanged. In other words, the block cannot be used to embed any secret bit. In addition, the proposed method records the ID number of the non-embeddable block to discriminate between the embeddable block and the non-embeddable block.

### 3.3. Extraction Stage

In the extraction stage, the secret data is extracted from the stego image T′. Moreover, T′ can be used to recover the original AMBTC decompressed image T. The details of the procedures are listed as follows:

1. Scan the stego AMBTC decompressed block $b\prime_{i}$ in T′ sequentially. If the difference between $H_{m}$ and $L_{m}$ is smaller than 4, then this block is considered a non-embeddable block. Otherwise, it is an embeddable block.

2. Retrieve the ID number of the selected dictionary D from the first embedded block. In the first embedded block, both the LSBs of the second *Hm* and the second *Lm* are extracted, i.e., binary representation of the ID number of the selected dictionary. Therefore, the proposed method can reconstruct the selected dictionary. In addition, both the LSBs are replaced by the first *Hm* and the first *Lm*, thereby recovering the original decompressed pixel.

3. Calculate the adjusting pixel values by using Pv=x′′i−Hm or Pv=x′′i−Lm for each embeddable block b′i. After obtaining Pv, we can look up the dictionary D to obtain the symbol Sp. After concatenating all Sp, we obtain the secret sequence S and recover the original AMBTC decompressed image T. The extraction and recovery pseudocode are shown in Algorithm 2.
**Algorithm 2:** Extraction and recovery pseudocodeforeach block b′i in T′ do  if Hm−Lm ≤4 then   /* non-embeddable block */    Do nothing;   else if bi is first embeddable block then    get the dictionary D number;  else    foreach pixel x′′i in b′i do      get the first Hm and Lm;      if x″i=Hm then       Pv=x′′i−Hm;      else       Pv=x′′i−Lm;      end      find the symbol Sp in D;      S=S+Sp;    end  endend

Figure 3 illustrates the extraction and recovery example. First, the dictionary is retrieved from the first embeddable block. Second, the adjusting pixel values are calculated as $P_{v}=x{\prime \prime }_{i}-H_{m}$ or $P_{v}=x{\prime \prime }_{i}-L_{m}$. Third, $P_{v}$ is mapped with the dictionary values to obtain $S_{p}$. Finally, $S_{p}$ is concatenated for obtaining the secret sequence S and the AMBTC decompressed block.

## 4. Experimental Results and Discussion

Some experimental cover images were tested to demonstrate the efficiency of the proposed scheme. In the experiments, the proposed scheme was verified using the following six test cover images: airplane, boat, lena, mandrill, peppers, and sailboat. As shown in Figure 4, all the images had the same size of 512 × 512 pixels with 256 grayscales, and the features of the images were diverse. The block size of the image presented in the AMBTC format was 4 × 4 pixels. A random binary sequence generated using a MATLAB (R2018a) function was used in the experiments as the secret sequence, where our secret data are the same as the secret data of the related works [15,16,17,19]. Note that each bit in the sequence has equal probability of being 0 or 1.

The proposed scheme was evaluated and compared with the aforementioned schemes in terms of two performance measures, i.e., hiding capacity and peak signal-to-noise ratio (PSNR). The hiding capacity can be defined as the number of secret data bits that can be hidden into a cover image. The PSNR is an objective measure used for determining the visual quality of an image. The higher the PSNR of a stego image, the better its visual quality is. The rule of thumb is that when the PSNR is higher than 30 dB, the human eyes cannot easily perceive the difference between the cover image and the stego image. PSNR is defined by
(10)$$PSNR=10\log_{10}\frac{255^{2}}{MSE},$$
(11)$$MSE=\;\frac{1}{M\times N}\sum_{i=1}^{M}\sum_{j=1}^{N}{\left(x_{ij}-x_{ij}^{\prime }\right)}^{2},$$
where $x_{ij}$ and *x*′*_ij_* are the original and stego grayscale pixel values located at (*i*, *j*), respectively. 

To present the superiority of the proposed scheme, we compared our scheme with the schemes presented by Li et al. [15], Lin et al. [16], Ou and Sun [17], and Malik et al. [19], as shown in Table 6. The proposed scheme achieved the highest data-hiding capacity for all five images except for the airplane image. The data-hiding capacity of the pixel value adjusting strategy was determined using the number of smooth blocks. If there are many smooth zones in a cover image, non-embeddable blocks are observed in abundance in the image. Moreover, the pixel value adjusting strategy used in the proposed scheme is modified at the most by 2, whereas the strategy used in the scheme proposed by Malik et al. is modified at the most by 1. Therefore, compared with the scheme presented by Malik et al., our scheme has a higher number of non-embeddable blocks in the airplane image. Non-embeddable blocks can be observed in black in Figure 5a,b. This is the main factor that causes the hiding capacity of the scheme proposed by Malik et al. to be better than that of the proposed scheme for the airplane image. For the other five images, the hiding capacity of our scheme is better than that of the scheme presented by Malik et al. by an enhancement value in the range of 10.13% to 29.89%. The hiding capacity of our scheme is better than the schemes proposed by Li et al., Lin et al., and Ou and Sun. Thus, we conclude that our scheme is better than the existing AMBTC- and BTC-based data-hiding schemes, in terms of the hiding capacity.

Table 7 lists the comparison between the method by Malik et al. and the proposed method in terms of structural similarity index (SSIM). As mentioned above, the SSIM value of the method by Malik et al. is greater than that of the proposed method because the proposed method embeds more secret data. In other words, the maximum hiding capacity of the proposed method is higher than that of the method by Malik et al.

The PSNR is the other factor for evaluating performance of a hiding scheme. Table 6 presents that the PSNR of our scheme is better than the schemes proposed by Li et al., Lin et al., and Ou and Sun. For the airplane stego image, the proposed scheme has a better PSNR but a weaker hiding capacity than the scheme proposed by Malik et al., because our scheme has a higher number of non-embeddable blocks than the scheme by Malik et al. Note that a non-embeddable block maintains the image quality but decreases the hiding capacity. For the other five stego images, the PSNR obtained using the proposed scheme is weaker than that obtained using the scheme proposed by Malik et al. However, because the PSNR difference is less than 0.29 dB for the five stego images, they would not be distinguishable by human vision due to such negligible differences. By contrast, the hiding capacities are significantly increased by a value in the range of 10.13% to 29.89% for the other five stego images. This implies that a tradeoff exists between the PSNR and hiding capacity when the pixel value adjusting strategy is used. The visual quality of the proposed scheme is observed to be above the average value of that of the baseline schemes.

## 5. Conclusions

A high-capacity data-hiding scheme was proposed in this study for an AMBTC-compressed image. The proposed scheme has many more properties than only high capacity. In this scheme, the dictionary-based coding scheme and the pixel value adjusting strategy were combined to increase the hiding capacity and attain a satisfactory visual quality. Experimental results reveal that the proposed scheme is better than the existing AMBTC-based data-hiding schemes in terms of the hiding capacity. Moreover, the visual quality of the proposed scheme is better than that of baseline schemes. In the future, we should combine the method by Liao et al. [20] with the proposed method to discriminate the image smoothness, thereby enhancing the hiding capacity. In addition, we should try to add the concept of partition strategy [21] into the proposed method to embed more secret data into the color images.

## Figures and Tables

**Figure 1 entropy-22-00145-f001:**
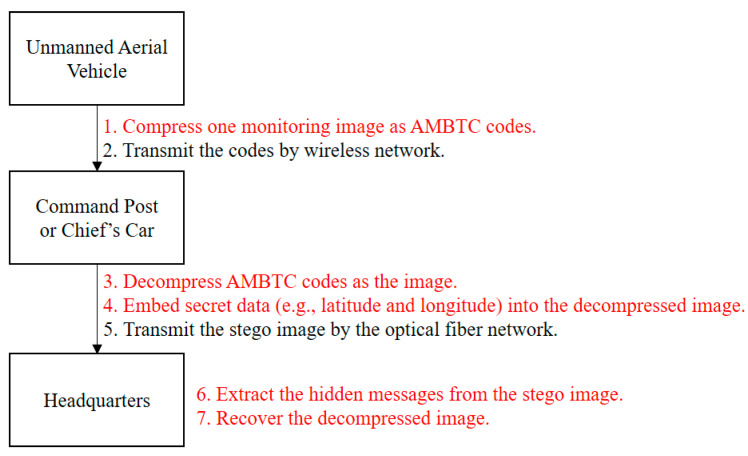
Flowchart of our applications.

**Figure 2 entropy-22-00145-f002:**
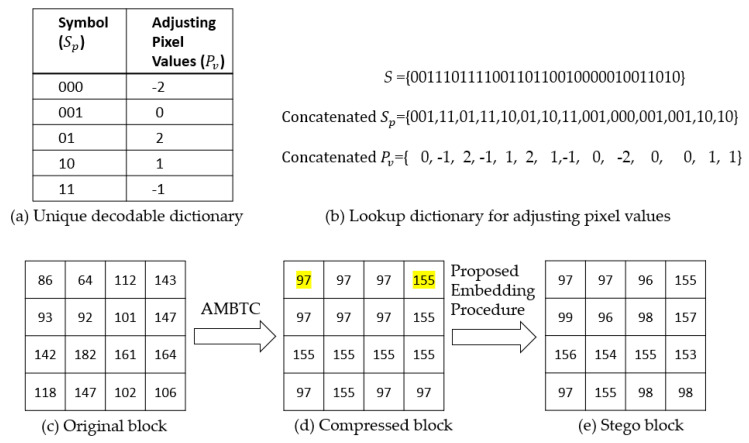
Example illustrating the proposed embedding stage.

**Figure 3 entropy-22-00145-f003:**
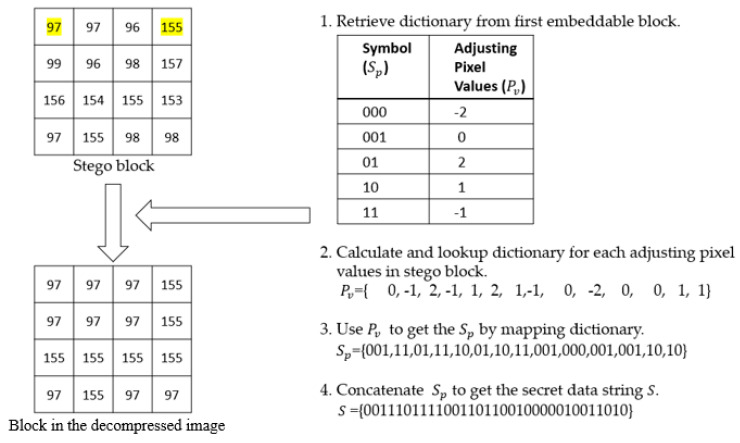
Example illustrating the proposed extraction stage.

**Figure 4 entropy-22-00145-f004:**
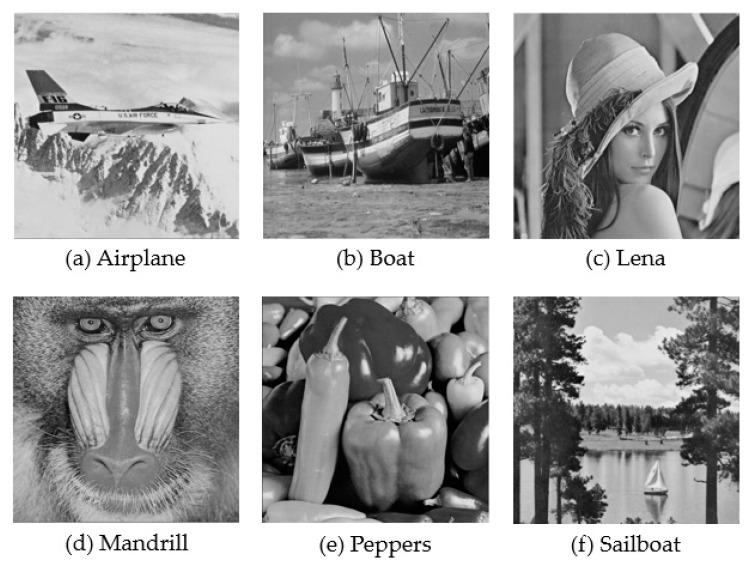
Test cover images.

**Figure 5 entropy-22-00145-f005:**
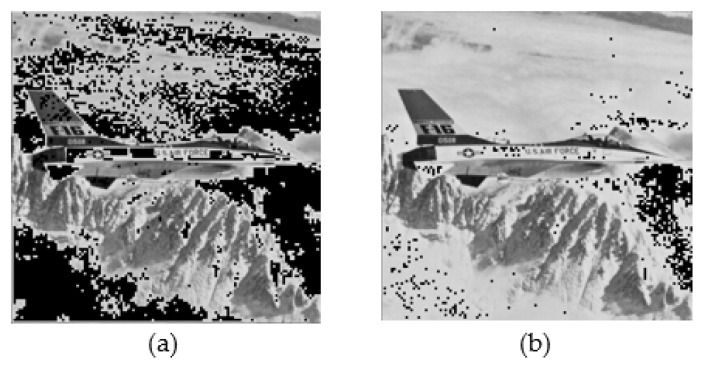
Non-embeddable blocks in “airplane.”: (**a**) Proposed scheme and (**b**) scheme proposed by Malik et al.

**Table 1 entropy-22-00145-t001:** Total number of data was 12.4670 with an entropy *H* of 2.1570 in the first dictionary $D_{1}$.

$\mathrm{Symbol}\;(S_{p})$	$\mathrm{Freq}.\;(S_{p}\;\mathrm{Count})$	Amount of Information
000	1	3.4594
001	4	1.4594
01	2	3.0444
10	4	2.0444
11	3	2.4594

**Table 2 entropy-22-00145-t002:** Total number of data was 12.1451 with an entropy *H* of 2.2264 in the second dictionary $D_{2}$.

$\mathrm{Symbol}\;(S_{p})$	$\mathrm{Freq}.\;(S_{p}\;\mathrm{Count})$	Amount of Information
00	5	1.7225
010	1	3.4594
011	2	2.4594
10	3	2.4594
11	4	2.0444

**Table 3 entropy-22-00145-t003:** Total number of data was 12.0751 with an entropy *H* of 2.2405 in the third dictionary $D_{3}$.

$\mathrm{Symbol}\;(S_{p})$	$\mathrm{Freq}.\;(S_{p}\;\mathrm{Count})$	Amount of Information
00	3	2.4150
01	3	2.4150
100	3	1.8301
101	1	3.4150
11	4	2

**Table 4 entropy-22-00145-t004:** Total number of data was 12.5602 with an entropy *H* of 2.1726 in the fourth dictionary $D_{4}$.

$\mathrm{Symbol}\;(S_{p})$	$\mathrm{Freq}.\;(S_{p}\;\mathrm{Count})$	Amount of Information
00	5	1.7225
01	3	2.4594
10	1	4.0444
110	3	1.8745
111	2	2.4594

**Table 5 entropy-22-00145-t005:** Dictionary example presenting the pixel value adjusting method.

Symbol (*S_p_*)	Freq. (S_p_ Count)	Sorted Index	Adjusting Pixel Values (*P_v_*)
000	1	5	−2
001	4	1	0
01	2	4	2
10	4	2	1
11	3	3	−1

**Table 6 entropy-22-00145-t006:** Comparison between hiding capacity and PSNR for different images for the proposed scheme and other AMBTC- and BTC-based schemes.

Method	Performance	Airplane	Boat	Lena	Mandrill	Peppers	Sailboat
Proposed	Hiding capacity (bits)	338,836	495,582	437,577	515,836	478,591	479,407
	PSNR (dB)	32.0908	31.0397	33.0562	26.9348	33.0087	29.7857
Malik et al. (2018)	Hiding capacity (bits)	397,147	397,380	397,348	397,105	397,057	397,466
	PSNR (dB)	31.9018	31.0926	33.102	26.9474	33.304	29.8081
Ou and Sun (2015)	Hiding capacity (bits)	223,039	217,264	234,004	141,919	238,969	219,169
	PSNR (dB)	30.71	29.54	30.87	26.02	31.59	28.61
Lin et al. (2013)	Hiding capacity (bits)	261,984	262,096	262,112	262,144	261,984	262,064
	PSNR (dB)	31.64	30.32	33.05	26.0047	32.2021	28.8049
Li et al. (2011)	Hiding capacity (bits)	17,659	16,580	16,789	16,880	17,264	16,990
	PSNR (dB)	30.18	31.83	31.05	27	32.35	28.43

**Table 7 entropy-22-00145-t007:** Comparison between hiding capacity and SSIM for different images for the proposed scheme and the other method by Malik et al.

Image	Malik et al.’s Method		Proposed Method	
	Hiding Capacity (bits)	SSIM	Hiding Capacity (bits)	SSIM
Airplane	397,147	0.947	338,836	0.9447
Boat	397,380	0.918	495,582	0.915
Lena	397,348	0.937	437,577	0.933
Mandrill	397,105	0.886	515,836	0.885
Peppers	397,057	0.931	478,591	0.927
Sailboat	397,466	0.915	479,407	0.912

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
