# Peer review of "High-Payload Data-Hiding Method for AMBTC Decompressed Images"

_entropy, 2020, doi:10.3390/e22020145_

Round 1
Reviewer 1 Report
Figure 1 (e) shows the stego block after hiding the secret bit.
Figure 1 (d) remains an AMBTC block.
However, Figure 1 (e) is no longer an AMBTC block after data hiding.
Therefore, it cannot be said that it is AMBTC based DATA HIDING method.
Therefore, it is difficult to ACCEPT this paper.
Author Response
Dear Editor-in-Chief,
Thank you very much for your letter of January 7, 2020, in which you noticed us that our manuscript entitled “High-Payload Data-Hiding Method for AMBTC Compression Images (Reference No. entropy-686847)” is required major revision for publication in Entropy.
The comments of the reviewers and associate editor were extremely helpful to us in preparing a clearer version. According to the reviewer’s valuable suggestions, we have modified the paper title and rewritten many paragraphs, where the new title is “High-Payload Data-Hiding Method for AMBTC Decompressed Images.” In addition, we have corrected all mistakes and added more explanation into the current manuscript. Thank you very much.
Attached are a copy of the revised version of the manuscript and a list of the revisions. Your acknowledgement will be greatly appreciated. Thank you very much again.
Sincerely yours,
Po-Liang Liu
Professor and Head
Graduate Institute of Precision Engineering,
National Chung Hsing University
Tel: +886-921-002-469
E-mail: pliu@dragon.nchu.edu.tw
Authors: Jung-Yao Yeh, Chih-Cheng Chen, Po-Liang Liu, Ying-Hsuan Huang
Title: High-Payload Data-Hiding Method for AMBTC Compression Images
Reference No. : entropy-686847
Comments
Response to the suggestions of Reviewer #1:
Figure 1 (e) shows the stego block after hiding the secret bit. Figure 1 (d) remains an AMBTC block. However, Figure 1 (e) is no longer an AMBTC block after data hiding. Therefore, it cannot be said that it is AMBTC based DATA HIDING method.Response: Thank you very much for your valuable comment. Yes, you’re right. The proposed method use AMBTC decompressed image rather than AMBTC block. Consequently, we have altered the paper title and revised the manuscript. The revised parts are as follows. Thank you very much again.
[MODIFIED]
(In Page 1) High-Payload Data-Hiding Method for AMBTC Decompressed Images (In Page 1) In this study, we proposed a high-capacity data-hiding scheme for AMBTC decompressed images. (In Page 2) To exploit the advantages of AMBTC compression, we proposed an AMBTC decompressed image based data-hiding scheme by using a pixel adjusting strategy. (In Page 4) In the scheme, secret data can be hidden into an AMBTC decompressed image. The AMBTC decompressed image can be losslessly reconstructed and the secret data can then be losslessly revealed from the reconstructed. (In Page 4) In other words, the proposed scheme uses the AMBTC decompressed image for embedding the secret data. (In Page 7) The AMBTC decompressed blocks in the original AMBTC decompressed image are sequentially scanned. (In Page 8) To embed these values, the original block must be compressed and decompressed using the AMBTC algorithm, as shown in Figure 2(c). After using the AMBTC algorithm, the AMBTC decompressed block can be reconstructed using a low mean value of 97 and a high mean value of 155, as shown in Figure 2(d). (In Page 8) For the AMBTC decompressed block, the pixel except the first and the first is increased by the adjusting pixel values to obtain the stego pixel. (In Page 8) Moreover, can be used to recover the original AMBTC decompressed image . (In Page 8) Scan the stego AMBTC decompressed block in sequentially. (In Page 8) After concatenating all , we will obtain the secret sequence and recover the original AMBTC decompressed image . (In Page 9) Finally, is concatenated for obtaining the secret sequence and the AMBTC decompressed block.Figure 2. Example illustrating the proposed embedding stage.
Please see the attached file.
Figure 3. Example illustrating the proposed extraction stage.
Please see the attached file.
We would like to thank you for being the reviewer of our manuscript. Your valuable comments are greatly helpful in making the manuscript more complete. We have done the revised work according to the reviewer’s comments.

Reviewer 2 Report
The manuscript has been improved and can be accepted after the following comments are adopted:
pseudocode is shown as follows: => pseudocode is shown in Algorithm 1https://ctan.org/pkg/algorithm2e Do not cite more than three references together.
schemes [1-24]. As for recent review on image data hiding, refer to http://dx.doi.org/10.1016/j.jisa.2019.102361 Change the caption of every table as one sentence.
http://abacus.bates.edu/~ganderso/biology/resources/writing/HTWtablefigs.html Embed each Algorithm as a text object instead of a lossy image. 5. Conclusions -> 5. Conclusion
Author Response
Dear Editor-in-Chief,
Thank you very much for your letter of January 7, 2020, in which you noticed us that our manuscript entitled “High-Payload Data-Hiding Method for AMBTC Compression Images (Reference No. entropy-686847)” is required major revision for publication in Entropy.
The comments of the reviewers and associate editor were extremely helpful to us in preparing a clearer version. According to the reviewer’s valuable suggestions, we have modified the paper title and rewritten many paragraphs, where the new title is “High-Payload Data-Hiding Method for AMBTC Decompressed Images.” In addition, we have corrected all mistakes and added more explanation into the current manuscript. Thank you very much.
Attached are a copy of the revised version of the manuscript and a list of the revisions. Your acknowledgement will be greatly appreciated. Thank you very much again.
Sincerely yours,
Po-Liang Liu
Professor and Head
Graduate Institute of Precision Engineering,
National Chung Hsing University
Tel: +886-921-002-469
E-mail: pliu@dragon.nchu.edu.tw
Authors: Jung-Yao Yeh, Chih-Cheng Chen, Po-Liang Liu, Ying-Hsuan Huang
Title: High-Payload Data-Hiding Method for AMBTC Compression Images
Reference No. : entropy-686847
Comments
Response to the suggestions of Reviewer #2:
The manuscript has been improved and can be accepted after the following comments are adopted:
pseudocode is shown as follows: => pseudocode is shown in Algorithm 1
Response: Thank you very much for your instruction and advice. We have rewritten the sentence. In addition, we have modified the similar sentence. Thank you very much again.
[MODIFIED]
(In Page 7) The embedding pseudocode is shown in Algorithm 1: (In Page 8) The extraction and recovery pseudocode is shown in Algorithm 2. Do not cite more than three references together. schemes [1-24].Response: Thank you very much for your valuable comment. We remove the unreasonable citation. The revised part is as follows. Thank you very much.
[MODIFIED]
(In Page 1) The schemes present for hiding data in an image can be broadly classified into two categories: irreversible data-hiding schemes [2]-[4] and reversible data-hiding schemes [5]-[7]. As for recent review on image data hiding, refer to http://dx.doi.org/10.1016/j.jisa.2019.102361Response: Thank you very much for your valuable comment. We have cited the paper. Thank you very much again.
[MODIFIED]
(In Page 1) In cryptography, e.g., Chaos-based encrypted systems, secure pseudo-random number generator and so on [1], users may be aware that there is an encrypted image, but they cannot efficiently decode the encrypted image unless they know the proper key. (In Page 12) Li C.; Zhang Y.; Xie E. Y. When an Attacker Meets a Cipher-image in 2018, A Year in Review. Journal of Information Security and Applications 2019, Volume 48, pp. 1-9. Change the caption of every table as one sentence.Response: Thank you very much for your advice. In each table, i.e., Table 1 through Table 4, we have merged two sentences as one sentence. The revised parts are as follows. Thank you very much.
[MODIFIED]
(In Page 6) Total number of data was 12.4670 with an entropy H of 2.1570 in the first dictionary . (In Page 6) Total number of data was 12.1451 with an entropy H of 2.2264 in the second dictionary . (In Page 6) Total number of data was 12.0751 with an entropy H of 2.2405 in the third dictionary . (In Page 6) Total number of data was 12.5602 with an entropy H of 1726 in the fourth dictionary . http://abacus.bates.edu/~ganderso/biology/resources/writing/HTWtablefigs.html Embed each Algorithm as a text object instead of a lossy image.Response: Thank you very much for your valuable comments. In the revised manuscript, we have regenerated losslessly images. The revised parts are as follows. Thank you very much again.
[MODIFIED]
(In Page 7) The embedding pseudocode is shown in Algorithm 1:Please see attached file. (In Page 8) The extraction and recovery pseudocode is shown in Algorithm 2.
Please see attached file “5. Conclusions -> 5. Conclusion”
Response: Thank you very much for your instruction. We have corrected the mistake. The revised part is as follows. Thank you very much again.
[MODIFIED]
(In Page 13) “5. Conclusion”We would like to thank you for being the reviewer of our manuscript. Your valuable comments are greatly helpful in making the manuscript more complete. We have done the revised work according to the reviewer’s comments.

Reviewer 3 Report
1. There are some errors on the paper, please correct the following:
- Equation (11) must include the value squared. See the following link https://es.mathworks.com/help/vision/ref/psnr.html
- "Based on the thumb rule of data encoding, ܵSp with the maximum occurrence frequency was encoded as the absolute minimum value. By contrast, ܵSp with the maximum occurrence frequency was encoded as the absolute maximum value". One of them is wrong.
2. In some cases, it is suggested that it be rewritten, as follows:
- Entropy (equation 8) as a negative value of the logarithm of probability. See the following link https://machinelearningmastery.com/what-is-information-entropy/
- The same comment for equation 7.
3. There are other metrics to evaluate the similarity between the original image and the stego image. For example the Structural Similarity Index (SSIM) for measuring image quality. See the link https://es.mathworks.com/help/images/ref/ssim.html. You should include this or similar metrics in your test.
4. Deep concerns, as follows:
- It's not clear from the motivation what the advantage is of compressing the image before applying the hiding process. It must be clarified.
- How do you get dictionaries? What is the reason for selecting the dictionary that provides the entropy of the small ones? Please answer the above questions both theoretically and experimentally.
- The results presented in Table 6 corresponded to how many experiments? Are they average values? In all cases the same string was used? How does the secret string influence the hiding capacity of the method?
Author Response
Dear Editor-in-Chief,
Thank you very much for your letter of January 7, 2020, in which you noticed us that our manuscript entitled “High-Payload Data-Hiding Method for AMBTC Compression Images (Reference No. entropy-686847)” is required major revision for publication in Entropy.
The comments of the reviewers and associate editor were extremely helpful to us in preparing a clearer version. According to the reviewer’s valuable suggestions, we have modified the paper title and rewritten many paragraphs, where the new title is “High-Payload Data-Hiding Method for AMBTC Decompressed Images.” In addition, we have corrected all mistakes and added more explanation into the current manuscript. Thank you very much.
Attached are a copy of the revised version of the manuscript and a list of the revisions. Your acknowledgement will be greatly appreciated. Thank you very much again.
Sincerely yours,
Po-Liang Liu
Professor and Head
Graduate Institute of Precision Engineering,
National Chung Hsing University
Tel: +886-921-002-469
E-mail: pliu@dragon.nchu.edu.tw
Authors: Jung-Yao Yeh, Chih-Cheng Chen, Po-Liang Liu, Ying-Hsuan Huang
Title: High-Payload Data-Hiding Method for AMBTC Compression Images
Reference No. : entropy-686847
Comments
Response to the suggestions of Reviewer #3:
The manuscript has been improved and can be accepted after the following comments are adopted:
Equation (11) must include the value squared. See the following link https://es.mathworks.com/help/vision/ref/psnr.html
Response: Thank you very much for your valuable advice. You’re right. We have corrected the equation. The revised equation is as follows. Thank you very much again.
[MODIFIED]
(In Page 10) Please see the attached file."Based on the thumb rule of data encoding, ܵSp with the maximum occurrence frequency was encoded as the absolute minimum value. By contrast, ܵSp with the maximum occurrence frequency was encoded as the absolute maximum value". One of them is wrong.
Response: Thank you very much for your valuable advice. We have corrected the sentence. The revised sentence is listed as follows. Thank you very much.
[MODIFIED]
(In Page 7) By contrast, with the lowest occurrence frequency was encoded as the absolute maximum value.In some cases, it is suggested that it be rewritten, as follows: Entropy (equation 8) as a negative value of the logarithm of probability. See the following link https://machinelearningmastery.com/what-is-information-entropy/. The same comment for equation 7.
Response: Thank you very much for your expert advice. It can effectively improve the format of manuscript. Consequently, we have modified the form of equations. The revised equations are as follows. Thank you very much again.
[MODIFIED]
(In Page 5) The amount of information in each symbol can be represented byPlease see the attached file.
Then, the average information per symbol interval is and can be represented by
Please see the attached file.
There are other metrics to evaluate the similarity between the original image and the stego image. For example the Structural Similarity Index (SSIM) for measuring image quality. See the link https://es.mathworks.com/help/images/ref/ssim.html. You should include this or similar metrics in your test.
Response: Thank you very much for your valuable comments. We have added the SSIM results into the manuscript. Thank you very much again.
[MODIFIED]
(In Page 11) Table 7 lists the comparison between Malik et al.’s method and the proposed method in terms of SSIM. As mentioned above, the SSIM value of the Malik et al.’s method is greater than that of the proposed method because the proposed method embeds more secret data. In other words, the maximum hiding capacity of the proposed method is higher than that of Malik et al.’s method.
Table 7. Comparison between hiding capacity and SSIM for different images for the proposed scheme and other Malik et al.’s method.
Image |
Malik et al.’s method |
Proposed method |
||
Hiding Capacity (bits) |
SSIM |
Hiding Capacity (bits) |
SSIM |
|
Airplane |
397,147 |
0.947 |
338,836 |
0.9447 |
Boat |
397,380 |
0.918 |
495,582 |
0.915 |
Lena |
397,348 |
0.937 |
437,577 |
0.933 |
Mandrill |
397,105 |
0.886 |
515,836 |
0.885 |
Peppers |
397,057 |
0.931 |
478,591 |
0.927 |
Sailboat |
397,466 |
0.915 |
479,407 |
0.912 |
Deep concerns, as follows: It's not clear from the motivation what the advantage is of compressing the image before applying the hiding process. It must be clarified.
Response: Thank you very much for your valuable comments. Fig. 1 shows the flowchart of our applications. First, one monitoring image on the unmanned aerial vehicle was compressed because the transmitting volume of wireless network is limited. When the command post or chief’s car receives the compression codes, they can decode as the decompressed image. In addition, they can embed secret data into the reconstructed image, thereby cheating hackers and avoiding attacks. Finally, the headquarters can extract secret data and recover the decompressed image. Thank you very much again.
[MODIFIED]
(In Page 4) Fig. 1 shows the flowchart of our application. First, one monitoring image on the unmanned aerial vehicle was compressed because the transmitting volume of wireless network is limited. When the command post or chief’s car receives the compression codes, they can decode as the decompressed image. In addition, they can embed secret data into the reconstructed image, thereby cheating hackers and avoiding attacks. Finally, the headquarters can extract secret data and recover the decompressed image.Fig. 1. Flowchart of our applications. Please see the attached file.
How do you get dictionaries? What is the reason for selecting the dictionary that provides the entropy of the small ones? Please answer the above questions both theoretically and experimentally.Response: Thank you very much for your valuable comments. The proposed method embedded the binary representation of the ID number of the selected dictionary into the least significant bit (LSB) of the second Hm and the second Lm. Note that the number of dictionaries is 4, thus the two LSBS can effectively represent the ID number.
In the extraction and recovery phase, both the LSBs of the second Hm and the second Lm are extracted, i.e., binary representation of the ID number of the selected dictionary. Therefore, the proposed method can reconstruct the selected dictionary. In addition, both the LSBs are replaced by the LSBs of the first Hm and the first Lm, thereby recovering the original decompressed pixel. The revised parts are as follows. Thank you very much again.
The following is used explain why use the dictionary of the smallest entropy. Assume that there is only one symbol’s type in the whole secret sequence. In other words, the other types never occur. In the case, the entropy is equal to 0, i.e., Afterwards, all symbols are replaced by the absolute minimum value “0”, thereby controlling the distortion level in the data embedding phase. Consequently, the proposed method selects the dictionary of the smallest dictionary.
[MODIFIED]
(In Page 7) In the first embeddable block, the binary representation of the ID number of the selected dictionary is embedded into the least significant bits (LSBs) of the second Hm and the second Lm. Note that the number of dictionaries is 4, thus the two LSBS can effectively represent the ID number. (In Page 8) Retrieve the ID number of the selected dictionary from the first embedded block. In the first embedded block, both the LSBs of the second Hm and the second Lm are extracted, i.e., binary representation of the ID number of the selected dictionary. Therefore, the proposed method can reconstruct the selected dictionary. In addition, both the LSBs are replaced by the first Hm and the first Lm, thereby recovering the original decompressed pixel. (In Page 5) The following explains why use the dictionary of the smallest entropy. Assume that there is only one symbol’s type in the whole secret sequence. In other words, the other types never occur. In the case, the entropy is equal to 0, i.e., Afterwards, the specific symbols are replaced by the absolute minimum value “0”, thereby controlling the distortion level in the data embedding phase. Consequently, the proposed method selects the dictionary of the smallest dictionary.The results presented in Table 6 corresponded to how many experiments? Are they average values? In all cases the same string was used? How does the secret string influence the hiding capacity of the method?
Response: Thank you very much for your valuable comments. Yes, our secret data are the same as the secret data of the related works. Consequently, the comparison in the manuscript is fair. The revised parts is as follow. Thank you very much again.
Regarding the hiding capacity, it becomes greater as the occurrence frequency of secret string of the three bits increases. The probability is about 25%. In other words, the proposed method can embed a trio of secret bits into one pixel with 25% probability.
[MODIFIED]
(In Page 10) A random binary sequence generated using a MATLAB function was used in the experiments as the secret sequence, where our secret data are the same as the secret data of the related works [15, 16, 17, 19].
We would like to thank you for being the reviewer of our manuscript. Your valuable comments are greatly helpful in making the manuscript more complete. We have done the revised work according to the reviewer’s comments.

Round 2
Reviewer 1 Report
The author has tried to improve the quality of the paper, including the title of the paper.
However, it does not match the basic concept of AMBTC.
In any case, AMBTC image must be the form of AMBTC type, after data hiding is applied. For example, if you changed Lena image to AMBTC, the concept of AMBTC should not change after data hiding. However, the author's suggestion changes the nature of AMBTC after hiding data. The authors, of course, say that the data was concealed after decompressing the AMBTC, so there is no problem. If that's the case, I wonder if there's a reason to use the AMBTC format.
Reviewer 3 Report
The paper has been improved. I suggest that the paper be accepted.
This manuscript is a resubmission of an earlier submission. The following is a list of the peer review reports and author responses from that submission.
Round 1
Reviewer 1 Report
In this paper, authors proposed a secret encoding algorithm and an embedding algorithm to hide secret data into the decompressed images. Experimental results showed that the hiding capacity of the proposed method is higher than that of Malik et al.'s method. However, there are some weaknesses in this paper, which are listed as follows. The paper should be improved significantly before acceptance. 1) Why cannot embed four secret bits into the decompressed pixels? 2) How to deal with the pixel with the overflow and underflow problems? 3) The 22th reference should be cited into the paper. And some related references should be added. [*]Data embedding in digital images using critical functions. Signal Processing: Image Communication, 2017, 58: 146-156 [*]A new payload partition strategy in color image steganography. IEEE Transactions on Circuits and Systems for Video Technology, 2019 4) This remainder -> The remainder 5) As for recent review on data hiding, refer to http://dx.doi.org/10.1016/j.jisa.2019.102361 6 ) Embed every equation among a text sentence. http://jmlr.csail.mit.edu/reviewing-papers/knuth_mathematical_writing.pdf 7) Put pseudocode in a separate Algorithm environment http://www.ctan.org/pkg/algorithm2eReviewer 2 Report
In page 8, generated Stego block (e) is not BTC, because a block is not composed of two quantization levels. As you know, BTC-based data hiding methods must remain BTC even after hiding data to it.